# An Altered Neurovascular System in Aging-Related Eye Diseases

**DOI:** 10.3390/ijms232214104

**Published:** 2022-11-15

**Authors:** Yoon Kyung Choi

**Affiliations:** Department of Bioscience and Biotechnology, Konkuk University, Seoul 05029, Republic of Korea; ykchoi@konkuk.ac.kr; Tel.: +82-2-450-0558

**Keywords:** retina, aging, neurovascular system, glaucoma, age-related macular degeneration, Alzheimer’s disease

## Abstract

The eye has a complex and metabolically active neurovascular system. Repeated light injuries induce aging and trigger age-dependent eye diseases. Damage to blood vessels is related to the disruption of the blood-retinal barrier (BRB), altered cellular communication, disrupted mitochondrial functions, and exacerbated aggregated protein accumulation. Vascular complications, such as insufficient blood supply and BRB disruption, have been suggested to play a role in glaucoma, age-related macular degeneration (AMD), and Alzheimer’s disease (AD), resulting in neuronal cell death. Neuronal loss can induce vision loss. In this review, we discuss the importance of the neurovascular system in the eye, especially in aging-related diseases such as glaucoma, AMD, and AD. Beneficial molecular pathways to prevent or slow down retinal pathologic processes will also be discussed.

## 1. Introduction

The hallmarks of aging include genomic instability, telomere attrition, epigenetic alterations, loss of proteostasis, deregulated nutrient sensing, mitochondrial dysfunction, cellular senescence, stem cell exhaustion, and altered intercellular communication [1]. In these processes, the vascular system plays a key role in cellular metabolism by supplying oxygen and nutrients [2]. Altered vascular systems in the eye (e.g., chronic hypoperfusion, inflammation, blood-retinal barrier (BRB) leakage, decreased nutrient supply, mitochondrial damage, immune cell infiltration, stem cell exhaustion, and altered intercellular communication) may facilitate the aging process [3,4,5,6]. Patients with glaucoma, age-related macular degeneration (AMD), and Alzheimer’s disease (AD) have altered microvascular networks in the retina compared with those in matched non-dementia controls [7,8] (Figure 1).

These changes in retinal microvasculature may reflect similar pathophysiological processes in the cerebral microvasculature of AD patients [9,10]. Peripapillary capillaries have been recognized as a highly specialized vasculature that supplies the nerve fiber layer. The retinal thickness in the peripapillary retinal nerve fiber layer is lower in glaucoma and AD patients than in healthy controls [8,11]. The short posterior ciliary artery has been found to exhibit transient vasospasm upon radical exposure in in vitro models [12], and reduced short posterior ciliary artery blood flow velocities are associated with glaucoma progression [13]. Retinal arteriolar narrowing has also been observed in patients [8]. Diminished vascular networks can affect neuronal survival. Retinal ganglion cells (RGCs) are lost by 25% near the fovea and in the nasal retina of aged individuals [14]. Thus, atrophy of the retina in aged patients may be involved in altered microvasculature that contributes to a reduced supply of O_2_ (hypoxia) and nutrients. This in turn leads to mitochondrial dysfunction, cellular senescence, stem cell exhaustion, and altered intercellular communication (Figure 2A,B). This review discusses the importance of the neurovascular system in the eye, especially in aging-related diseases such as glaucoma, AMD, and AD. Beneficial molecular pathways to prevent or delay retinal pathologic processes are also discussed.

## 2. BRB Dysfunction in Aging and Diseases

The BRB consists of the inner and outer BRB [15]. The inner BRB is formed by tight junctions between retinal capillary endothelial cells, while the outer BRB is formed by tight junctions between retinal pigment epithelial (RPE) cells [16]. Glaucoma and AD are associated with inner BRB breakdown in the retina, while AMD is closely related to outer BRB breakdown (Figure 2C). At the early pathological stage of aging-related diseases, the retinal capillary degeneration and compromised BRB integrity may provide important clues for diagnosis and therapy [10]. AD brains show leakage of the cerebral capillary endothelium [17], indicating abnormalities in the blood–brain barrier (BBB). The albumin ratio as a marker of BBB permeability correlates with the severity of dementia [18]. In addition to abnormalities in the BBB, inner BRB disruption and retinal capillary degeneration were also detected in murine AD models [10]. Along with capillary degeneration, pericyte deficiency has been observed in the retina of AD transgenic mice [10]. RPE cells play an important role in immune regulation because aging RPE becomes immunologically active for immune cell infiltration into retinal neurons through the damaged outer BRB [19]. Oxidative stress affects both the inner and outer BRB. Reactive oxygen species (ROS) and reactive nitrogen species (RNS) reduce tight junctions between endothelial and RPE cells [20,21,22].

## 3. Age-Related Retinal Diseases

Age-related neurodegenerative eye diseases, including glaucoma, AMD, and AD, are characterized by the accelerated loss of retinal neurons and their axons. These diseases are interrelated and share common molecular mechanisms induced by repeated light injury and the consequent overexpression of oxidative stress. The prevalence and incidence of primary open-angle glaucoma exponentially increase with age [23]. The global incidence of glaucoma has reached approximately 79 million in 2020, and this number is expected to increase to over 111 million by 2040. Glaucoma involves damaged optic nerves, loss of RGCs by apoptosis, and altered connection between RGCs and the visual cortex [24]. RGCs have a limited capacity for regeneration following damage in adulthood [25]. Thus, glaucoma-induced loss of RGCs may be irreversible in aged patients.

AMD has affected 196 million individuals aged 45–85 years worldwide in 2020, accounting for approximately 8.7% of the population. The majority (>85%) of AMD patients have the dry form of the disease, which is characterized by extracellular deposits called drusen (heterogeneous debris, including lipid accumulation between the RPE and Bruch’s membrane) beneath the RPE and subsequent RPE atrophy in the macula. However, there is currently no treatment for this AMD type. In dry AMD, drusen, choroidal ischemia, and vitreoretinal adhesion are independently determined by genetics and environment (i.e., smoking and diet) and may occur concurrently in variable proportions. If the resulting hypoxia and consequent vascular endothelial growth factor (VEGF) accumulation exceed the threshold, this will trigger imbalanced choroidal neovascularization [26] (Figure 3). Meanwhile, wet AMD patients, accounting for approximately 15% of the total AMD patients, are administered anti-VEGF antibody therapy to inhibit blindness [27,28]. Narrowing vessels result in chronic hypoxia in endothelial cells with age. Chronic hypoxia and consequent accumulation of hypoxia inducible factor (HIF)-α (e.g., HIF-1α and HIF-2α) in RPE cells may be central AMD risk factors showing Bruch’s membrane thickening and metabolic changes (i.e., lipid accumulation, VEGF upregulation) in RPE limit glucose delivery to photoreceptors [29].

Approximately 1 in 9 individuals aged ≥ 65 years have AD. Dementia has already affected 55 million individuals aged ≥ 65 years in 2022 [30]. Narrowing vessels and prolonged hypoperfusion alter waste disposal systems in aging and diseases, leading to reduced clearance of aggregated and misfolded proteins and BRB disruption [9,10,31]. Chronic cerebral hypoperfusion causes significant cognitive decline concurrent with increased levels of tau phosphorylation, dysregulated synaptic proteins, and altered mitochondrial ultrastructure in neuronal cells [32,33,34]. Chronic overexpression of heme oxygenase-1 (HO-1) produces labile iron [35], and HO-1 can be induced by hypoxia [36]. Excessive intracellular labile iron levels lead to ferroptosis and consequent cell death, and these mechanisms underline the pathology of several neurodegenerative diseases [37]. Endothelial cell ferroptosis also triggers inflammatory responses through the NOD-like receptor family pyrin domain-containing 3 (NLRP3) [38]. As lipid peroxidation and accumulation can be related to drusen deposition, ferroptosis may be a critical regulator of age-related eye diseases [39].

## 4. Retinal Cells in Aging and Diseases

Chronic hypoxia and the accumulation of toxic agents mediate inflammasome formation and ferroptosis in endothelial cells, consequently inducing insufficient nutrient and oxygen supply to the surrounding neurovascular unit. In this section, we discuss the various neurovascular cells that affect endothelial cells.

### 4.1. Stem Cells

Stem cell exhaustion has been observed during aging. Healthy vessels and intact pericytes are important for stem cell proliferation and differentiation [31,40,41]. In the retina, stem cells can be generated from multipotent progenitor cells and Müller glial cells [42]. During embryonic development of the eye, a group of founder cells in the optic vesicle gives rise to multipotent progenitor cells that generate all neurons and glia of the mature retina. In most vertebrates, a small group of retinal stem cells persist at the margin of the retina near the junction with the ciliary epithelium [43]. In a fish model, multipotent adult retinal stem cells differentiated into various retinal neurons and glia and formed an arched-continuous stripe by clone transplantation at the blastula stage (i.e., the early stage in embryogenesis) [44]. By applying tools available to fish, retinal stem cells can be deciphered based on their localization, growth, and differentiation [44,45].

Boosting RGC regeneration may be a potential therapeutic strategy for glaucoma. Human pluripotent stem cells are attractive candidates for translational approaches because of their ability to divide and differentiate into RGCs [46]. Human pluripotent stem cell-derived retinal organoids may serve as useful models for RGC development [47]. Retinal organoid-derived RGCs actively extend neurites in the presence of Netrin-1 (a chemotropic factor) or brain-derived neurotrophic factor (BDNF) [47]. In zebrafish, Müller glial cells act as radial glial-like neural stem cells and generate rod progenitors [42]. During injury, Müller glial cells can stimulate adult neurogenesis, partly through epigenetic changes [48]. The nuclei of Müller glial cells translocate to the apical surface and divide asymmetrically to give rise to proliferating multipotent retinal progenitors that accumulate around the radial glial fiber and migrate to the appropriate retinal laminae to regenerate neurons [42]. In age-related diseases, stem cell extinction can be accelerated by neurovascular cell miscommunication because vascular cells, such as endothelial cells and pericytes, are damaged or dead.

### 4.2. Retinal Pigment Epithelium

The RPE is a monolayer of cells that underlie and support photoreceptors in the retina. RPE plays two critical roles in the function of retinal photoreceptors. First, the membranous disks in the outer segment, which house the light-sensitive photopigment and outer proteins involved in phototransduction, are turned over within approximately 12 days. New outer segment disks are continuously formed near the base of the outer segment, whereas the aged portion of the disks is eliminated. During their lifespan, disks move gradually from the base of the outer segment to the tip to remove expended receptor disks. This shedding involves “pinching off” of a clump of receptor disks by the outer segment membrane of the photoreceptor [49]. This enclosed clump of disks that may be exposed to photo-oxidative damage is phagocytosed by the RPE [49]. During aging, the RPE undergoes significant morphological and functional changes. The number of RPE declines and the size increases with decreases in phagocytic and lysosomal activities [50,51]. Second, RPE regenerates photopigment molecules (i.e., melanin) after exposure to light, and this effect is reduced with aging [52]. Photopigment is cycled continuously between the outer segment of the photoreceptor and RPE. Disruption of cell–cell interactions between the RPE and retinal photoreceptors has severe consequences on vision.

Endogenous regeneration of the RPE has been reported. Injury-adjacent RPE cells proliferate and differentiate into RPE cells in zebrafish [53]. In addition, quiescent human RPE stem cells have been identified, and adult RPE stem cells can proliferate in vitro and differentiate into RPE or mesenchymal cell types [54]. As RPE is susceptible to ROS/RNS, the antioxidant milieu of the eye may be beneficial for RPE stem cells.

### 4.3. Glia

Glial cells are a complex population of cells expressing different transcription factors and neurotrophic factors in different environments [55,56,57,58]. Aging glial cells, such as astrocytes, Müller glial cells, oligodendrocytes, and microglia, can be involved in uncontrolled inflammation and impaired cell–cell networks [55]. Aged astrocytes can no longer support neuron-oligodendrocyte interactions [59]. Similar to microglia, astrocytes are also involved in eliminating neurons by phagocytosis [60]. Autophagy-dysregulated astrocytes are observed in the aging hippocampus [61]. Autophagy is an intracellular degradation process, and reduced autophagy in the aged retina causes accumulation of damaged components [6,62]. Aged astrocytes and microglia undergo morphological alterations, accumulation of autophagosomes, and impaired photoreceptor degradation [63,64,65].

Aging is also characterized by gliosis, loss of axons, demyelination, and vision loss. Gliosis is a reactive process that includes the proliferation of glial cells, such as astrocytes, after injury. Astrocytes express glial fibrillary acidic protein (GFAP) under physiological conditions, and Müller glial cells express GFAP in RPE cells during AMD [66]. In animal models, glaucomatous optic nerve injury triggers reactive astrocytes and axonal degeneration [67]. These are followed by microglia activation, modest loss of oligodendrocytes, and consequent demyelination [67].

Proper intercellular interactions can be important to delay the aging process. Given that glial cells comprise the neurovascular unit linking endothelial cells and neurons, disruption of these cellular interactions can exaggerate neurovascular miscommunications. The astrocytic water channel aquaporin-4 is densely expressed by astrocytes almost exclusively at the end-feet; however, aquaporin-4 loses its polarization in reactive astrocytes and is found to be diffusively expressed [5,68]. Glia-cell-derived VEGF and its receptor VEGFR2 expressed in endothelial cells stimulate the survival of endothelial cells and angiogenesis [29,69,70,71], which are reduced in aging [72]. Müller glial cells can inhibit excessive retinal endothelial cell proliferation by upregulating transforming growth factor β2 [73]. Thus, morphological and functional changes in glia affect the neurovascular system under pathological conditions with aging.

### 4.4. Pericytes

Pericytes are vulnerable to ischemic conditions [40]. Aging retina with chronic ischemia reduces the ability of pericytes to relax after constriction, leading to a further decrease in blood flow [74,75]. In the aged rat retina, interactions between pericyte and endothelial cells become weak and disrupted [76]. A substantial vascular pericyte deficiency, along with prominent vascular Aβ deposition, was detected in the retina of AD (APP_SWE_/PS1_ΔE9_) mice, and this was inversely correlated with the extent of degenerated capillaries [10]. Angiopoietin-1 is expressed in retinal pericytes, and its receptor Tie2 is expressed in retinal endothelial cells [77]. Interactions between pericytes and endothelial cells through the angiopoietin-1–Tie2 pathway promote angiogenesis and protect retinal neurons during ischemic injury [77]. However, the interaction between pericytes and endothelial cells is weakened during aging, resulting in loss of capillary coverage, distorted retinal vessels, and breakdown of BRB foci [10,76] and possibly leading to impaired exchange of metabolites required for optimal neurovascular function. Moreover, pericytes respond to carbon monoxide (CO), nitric oxide (NO), and adenosine triphosphate (ATP) [78,79] and may communicate with neural stem cells, endothelial cells, and photoreceptors. Reduced capillary diameter and impaired blood flow at pericyte locations in eyes correlate with high intraocular pressure [80]. Considering the critical role of pericytes in ocular perfusion and blood flow in aged retinas, protecting healthy pericytes from age-related damage may be crucial for maintaining healthy vision.

## 5. Metabolic Disturbances in the Retina

Adequate evidence supports that metabolic disruptions are closely related to eye aging. The retina requires large amounts of ATP through mitochondrial functions for phototransduction as well as for maintaining a depolarized state in the absence of light, leading to the induction of chronic hypoxia. Repeated light injury and ischemic stress triggers ROS/RNS production and uncontrolled inflammatory responses. The retina is an immune-privileged tissue that is highly sensitive to inflammatory damage. Aging-related reductions in cellular defense mechanisms against inflammation make the retina vulnerable to such damage.

### 5.1. Mitochondria

Age-related metabolic dysfunction may play a key role in the etiology of neurodegenerative eye diseases [81]. Mitochondria are involved in ATP generation through oxidative phosphorylation (OXPHOS) and regulate cell death through apoptosis. Age-related mitochondrial damage and decreased ATP production have been reported [6]. Glaucoma pathology is related to the apoptosis of RGCs [82], implying that malfunctional mitochondria underlie its pathogenesis. In an animal model of glaucoma, glucose transporter levels were markedly lower in GFAP-positive retinal glial cells than in normal control cells [83]. The retina obtained from the glaucoma-induced model showed age-related decline in the NAD^+^/NADH ratio and expression of peroxisome proliferator-activated receptor γ coactivator 1α (PGC-1α) [83]. Thus, glaucoma patients may have a deficit in the generation of new mitochondria. Mitochondrial DNA is deleted in human RPE tissues from donors aged 60–110 years [4]. Patients with AD also show diminished mitochondrial density and area within the cerebral capillary endothelium [17]. Postmortem brains from AD patients demonstrate a mitochondria-on-a-string phenotype (mitochondrial fission arrest) in the hippocampus and entorhinal cortex [84]. This altered mitochondrial phenotype is mimicked in young (10 weeks) wild-type mice exposed to acute hypoxic conditions [84].

Imports of nuclear-encoded proteins into the mitochondria through translocase of the outer membrane are vital for mitochondrial functions [85]. Analysis of mitochondria-enriched homogenates from the postmortem neocortex of AD and age-matched controls revealed that Tom20 and Tom70 expression was reduced in AD [86]. Tom20 and Tom22 may play key roles in ATP production and OXPHOS in astrocytes [87]. Thus, nucleus-mediated protein import into the mitochondria is critical for energy supply. Altered morphological feature of ferroptosis is mitochondrial dysfunction, including a smaller mitochondrial volume through imbalanced fission, fusion, and rupture of the mitochondrial outer membrane [88]. Iron accumulation and lipid peroxidation in the aging retina may be attributed to ferroptosis-induced retinal cell death in age-related retinal diseases [39]. Thus, aging may facilitate mitochondrial depletion and consequently diminish the interactions between the nucleus and mitochondria (Figure 4).

### 5.2. Cellular Senescence

Cellular senescence is closely related to mitochondrial functional proteins such as AMP-activated protein kinase α (AMPKα), nicotinamide phosphoribosyltransferase, and sirtuins (SIRTs) [89]. AMPKα activation is required for the protection of photoreceptors and RPE from acute injury and delayed inherited retinal degeneration [90]. Protective mechanisms may include decreased oxidative stress, reduced DNA damage, and increased mitochondrial biogenesis [90]. SIRTs (SIRT1, SIRT2, SIRT3, SIRT4, SIRT5, SIRT6, and SIRT7) function as NAD^+^-dependent protein deacetylases [91]. SIRTs, except SIRT5, are expressed in the human retina [92]. Loss of Nampt in aging RPE cells reduces NAD^+^ availability and SIRT1 expression, thereby facilitating cellular senescence [93]. SIRT1 deacetylases PGC-1α, leading to increased PGC-1α activity, such as that seen in mitochondrial biogenesis [94]. PGC-1α protects the RPE of the aging retina against oxidative stress-induced degeneration through the regulation of senescence and mitochondrial quality control [95]. In an optic nerve crush model, SIRT1 overexpression in RGCs reduces RGC loss, thus preserving visual ability [96]. Identifying additional molecular mechanisms is crucial for determining cellular senescence pathways in the eyes.

### 5.3. Inflammation

In inflammatory responses, nuclear factor kappa light chain enhancer of activated B cells are a critical transcription factor for the expression of various genes, including NLRP3, inducible NOS, and immune cell attractants (i.e., intercellular adhesion molecule 1 and vascular cell adhesion molecule 1) [97,98]. Ferroptosis and ROS/RNS production trigger inflammatory responses via the NLRP3 inflammasome activation, which is related to tau pathology [38,99]. NLRP1- and NLRP3-mediated inflammasome activation induces caspase-1-induced neuronal pyroptosis in the retina during ocular hypertension injury [100]. In AMD, NLRP3-mediated cell death mechanisms (i.e., pyroptosis) underlie RPE degeneration [101]. NLRP3 deficiency reduces the number of VEGF-A-induced choroidal neovascularization lesions and RPE barrier breakdown [102]. VEGF-A overexpression mice show NLRP3 inflammasome activation in RPE cells [102]. Thus, the crosstalk between VEGF-A and NLRP3 triggers age-dependent progressive AMD in vivo.

The role of autophagy in NLRP3-mediated inflammasome activation has been previously reported. Autophagy induced by inflammasomes may reduce inflammasome activity, possibly due to sequestration and subsequent degradation of excessive cytokine precursors, such as pro-interleukin-1β in the autophagosome [103,104]. Thus, controlling excessive inflammation via autophagy may protect the eyes from neurovascular disruption.

## 6. Therapeutic Approaches

Oxidative stress-reducing agents may exert anti-aging effects in the retina. In endothelial cells, appropriate levels of VEGF and endothelial nitric oxide synthase (eNOS)/NO can maintain endothelial cell survival [105] and regulate intraocular pressure by dilating microvessels [106]. The NO-mediated guanylate cyclase/cGMP pathway increases ocular blood flow and confer neuroprotection [106]. In addition, the eNOS/NO pathway improves vessel health partly through crosstalk with HO-1/CO [107], leading to vasodilation and anti-inflammation. Nuclear factor erythroid-2-related factor 2 (Nrf2) is a transcription factor for HO-1 [36]. In a rat model of glaucoma, activation of the Nrf2/HO-1 pathway protected RGCs from chronic ocular hypertension [108]. Nrf2-deficient mice show age-related drusen-like deposits, accumulation of lipofuscin, spontaneous choroidal neovascularization, and sub-RPE deposition of inflammatory proteins after 12 months [109]. Some of these features, such as drusen formation, are hallmarks of AMD [109]. Nrf2-deficient RPE show increased proportions of autophagosomes, autolysosomes, swollen mitochondrial fragments next to autophagic vacuoles, undigested photoreceptor outer segments, and lipofuscin [109]. Healthy mitochondrial function is critical against retinal aging because the eyes are highly metabolic tissues. In a study of oxidative stress astrocyte glial cells, Korean red ginseng upregulated Nrf2/HO-1 pathways, leading to increased levels of OXPHOS and cytochrome c and Tom20-mediated mitochondrial membrane potential cells [110]. In astrocytes, Korean red ginseng-induced Tom20 expression is mediated by the upregulation of Nrf2/HO-1-mediated SIRT1, SIRT2, and SIRT3 [110]. A ketogenic diet may help protect the retina from chronic stress by enhancing mitochondrial activity [83]. Particularly, a ketogenic diet increases the levels of Nrf2, HO-1, and BDNF in the retina under chronic metabolically stressed optic nerves [83].

In addition to anti-inflammation for neuroprotection, one strategy for the repair of retinal neurons is stem cell-based regeneration. Müller cells can differentiate into cells that resemble pluripotent stem cells. As stem cells, Müller glial cells generate new neurons and glial cells, including Müller cells [42]. Maintaining the functions of Müller glial cells is important because they protect photoreceptors through the release of neurotrophic factors, such as BDNF [111]. Adult hippocampal neurogenesis combined with BDNF upregulation improves cognition in AD mouse models [112], similar to the beneficial effects of exercise [112,113]. Overall, several pathways may be required for anti-aging and neuroprotection, and these pathways include anti-inflammation, mitochondrial activity, stem cell-based regeneration, and cell–cell communication.

## Figures and Tables

**Figure 1 ijms-23-14104-f001:**
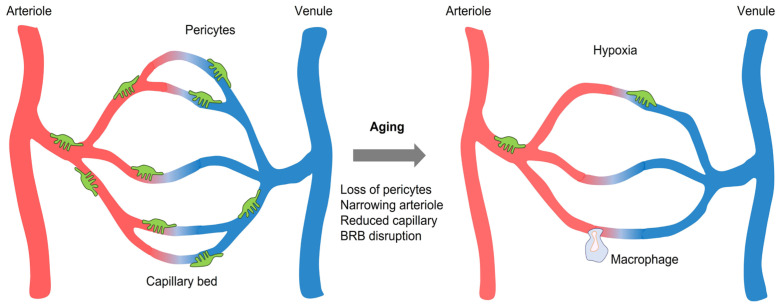
Comparison between the young and old vessels in the retina. Aging results in narrowing arteriole, retinal capillary degeneration, pericyte depletion, hypoxia, and BRB disruption. *Abbreviation*. BRB, blood-retinal barrier.

**Figure 2 ijms-23-14104-f002:**
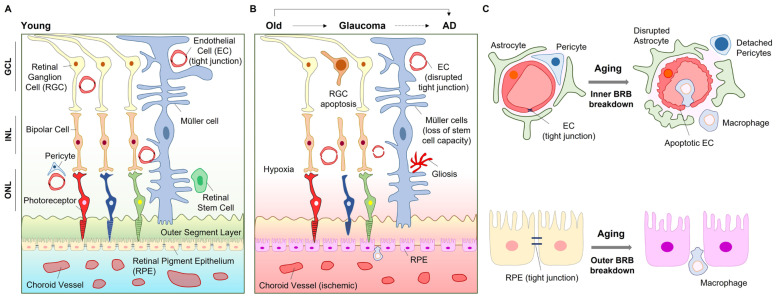
Aging processes in the eye. Young retina shows intact cellular communications for fine vision (**A**); however, aging and consequent age-related eye diseases (i.e., glaucoma, AD) reduce neurovascular cell communication in the eye (**B**). Aging and age-dependent eye diseases show inner BRB breakdown and outer BRB breakdown (**C**). *Abbreviation*. AD, Alzheimer’s disease; GCL, ganglion cell layer; INL, inner nuclear layer; ONL, outer nuclear layer.

**Figure 3 ijms-23-14104-f003:**
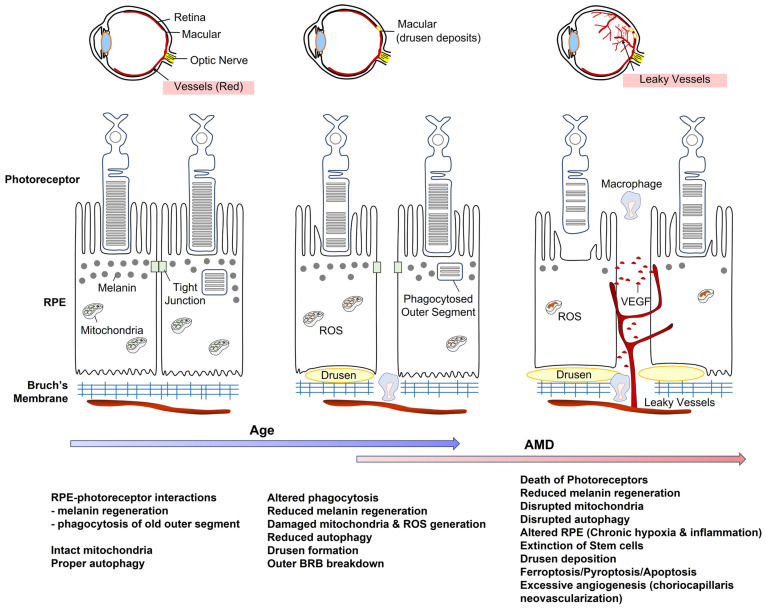
Intercellular communication between RPE and photoreceptor can be diminished by aging. This effect can be further exaggerated by the AMD process. *Abbreviation*. AMD, age-related macular degeneration; ROS, reactive oxygen species; RPE, retinal pigment epithelium, VEGF, vascular endothelial growth factor.

**Figure 4 ijms-23-14104-f004:**
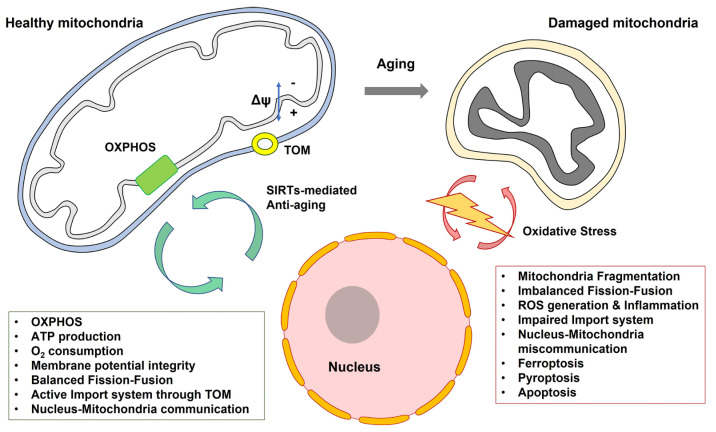
Communication between mitochondria and nucleus can be diminished by aging. *Abbreviation*. OXPHOS, oxidative phosphorylation; TOM, the translocase of the outer membrane.

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
