# Peer review of "An Altered Neurovascular System in Aging-Related Eye Diseases"

_ijms, 2022, doi:10.3390/ijms232214104_

Round 1

Reviewer 1 Report

The authors of the manuscript entitled “An Altered Neurovascular System in Aging-related Eye Diseases” discuss the importance of the neurovascular system in the eye, especially in aging-related diseases such as glaucoma, AMD, and AD.

The manuscript is clearly written and is well suited for the prepared special issue, “Molecular Mechanisms of Angiogenesis in Health and Diseases 2.0”.

I only have a few minor comments and remarks:

1/ In the introduction, can the authors better explain, why they focus on the three diseases mentioned. Are these diseases the most common or best described?

2/ It must always be clear from the text whether authors present results of the analysis of human samples or the results obtained using model organisms. Most of the text is fine in this respect, however, e.g. the first paragraph in section 4.3. Glia is not entirely clear.

Author Response

Reviewer #1

The authors of the manuscript entitled “An Altered Neurovascular System in Aging-related Eye Diseases” discuss the importance of the neurovascular system in the eye, especially in aging-related diseases such as glaucoma, AMD, and AD.

The manuscript is clearly written and is well suited for the prepared special issue, “Molecular Mechanisms of Angiogenesis in Health and Diseases 2.0”.

I only have a few minor comments and remarks:

  1. In the introduction, can the authors better explain, why they focus on the three diseases mentioned. Are these diseases the most common or best described?

Response: As far as I know, these three diseases (i.e., glaucoma, age-related macular degeneration, and Alzheimer’s disease) are closely related to aging processes. Since these diseases are most age-related diseases, I chose them.

  1. It must always be clear from the text whether authors present results of the analysis of human samples or the results obtained using model organisms. Most of the text is fine in this respect, however, e.g. the first paragraph in section 4.3. Glia is not entirely clear.

Response: Per your valuable comment, I reorganized 4.3. Glia section in Line 184-212.

4.3. Glia

Glial cells are a complex population of cells expressing different transcription factors and neurotrophic factors in different environments [55-58]. Aging glial cells, such as astrocytes, Müller glial cells, oligodendrocytes, and microglia, can be involved in uncontrolled inflammation and impaired cell-cell networks [55]. Aged astrocytes can no longer support neuron-oligodendrocyte interactions [59]. Similar to microglia, astrocytes are also involved in eliminating neurons by phagocytosis [60]. Autophagy-dysregulated astrocytes are observed in the aging hippocampus [61]. Autophagy is an intracellular degradation process, and reduced autophagy in the aged retina causes accumulation of damaged components [6, 62]. Aged astrocytes and microglia undergo morphological alterations, accumulation of autophagosomes, and impaired photoreceptor degradation [63-65].

Aging is also characterized by gliosis, loss of axons, demyelination, and vision loss. Gliosis is a reactive process that includes the proliferation of glial cells, such as astrocytes, after injury. Astrocytes express glial fibrillary acidic protein (GFAP) under physiological conditions, and Müller glial cells express GFAP in RPE cells during AMD [66]. In animal models, glaucomatous optic nerve injury triggers reactive astrocytes and axonal degeneration [67]. These are followed by microglia activation, modest loss of oligodendrocytes, and consequent demyelination [67].

Proper intercellular interactions can be important to delay the aging process. Given that glial cells comprise the neurovascular unit linking endothelial cells and neurons, disruption of these cellular interactions can exaggerate neurovascular miscommunications. The astrocytic water channel aquaporin-4 is densely expressed by astrocytes almost exclusively at the end-feet; however, aquaporin-4 loses its polarization in reactive astrocytes and is found to be diffusively expressed [5, 68]. Glia-cell-derived VEGF and its receptor VEGFR2 expressed in endothelial cells stimulate the survival of endothelial cells and angiogenesis [29, 69-71], which are reduced in aging [72]. Müller glial cells can inhibit excessive retinal endothelial cell proliferation by upregulating transforming growth factor b2 [73]. Thus, morphological and functional changes in glia affect the neurovascular system under pathological conditions with aging.

Reviewer 2 Report

The authors reviewed the molecular mechanism related to aging-related eye diseases. It is a well-written article to have worth being published in Molecular Sciences, with a minor revision.

The authors summarized the beneficial molecular pathways and therapeutical approaches. However, it is not clear how the two topics are merged together. The reviewer especially failed to understand the concluding remark, saying "these interactions and communications may be enhanced by exercise, balanced diet, non-smoking, and minimal light injury."

I am not sure whether the authors have any scientific evidence to support this statement. The section on therapeutic approaches is quite interesting, but the authors need to be improved.

Author Response

Reveiwer #2

The authors reviewed the molecular mechanism related to aging-related eye diseases. It is a well-written article to have worth being published in Molecular Sciences, with a minor revision.

The authors summarized the beneficial molecular pathways and therapeutical approaches. However, it is not clear how the two topics are merged together. The reviewer especially failed to understand the concluding remark, saying "these interactions and communications may be enhanced by exercise, balanced diet, non-smoking, and minimal light injury."

I am not sure whether the authors have any scientific evidence to support this statement. The section on therapeutic approaches is quite interesting, but the authors need to be improved.

Response: To improve the section, I decided to remove one sentence ‘These interactions and communications may be enhanced by exercise, balanced diet, non-smoking, and minimal light injury.’ As your valuable comment, further study may need to support whether protective lifestyles can impact cellular communications.
